# AUTOFORMALIZER WITH TOOL FEEDBACK

## ABSTRACT

Autoformalization addresses the scarcity of data for Automated Theorem Proving (ATP) by translating mathematical problems from natural language into formal statements. Efforts in recent work shift from directly prompting large language models to training an end-to-end formalizer model from scratch, achieving remarkable advancements. However, existing formalizer still struggles to consistently generate valid statements that meet syntactic validity and semantic consistency. To address this issue, we propose the Autoformalizer with Tool Feedback (ATF), a novel approach that incorporates syntactic and consistency information as tools into the formalization process. By integrating Lean 4 compilers for syntax corrections and employing a multi-LLMs-as-judge approach for consistency validation, the model is able to adaptively refine generated statements according to the tool feedback, enhancing both syntactic validity and semantic consistency. The training of ATF involves a cold-start phase on synthetic tool-calling data, an expert iteration phase to improve formalization capabilities, and Direct Preference Optimization to reduce ineffective revisions. Experimental results show that ATF markedly outperforms a range of baseline formalizer models, with its superior performance further validated by human evaluations. Subsequent analysis reveals that ATF demonstrates excellent inference scaling properties. Moreover, we open-source Numina-ATF, a dataset containing 750K synthetic formal statements to facilitate advancements in autoformalization and ATP research.

## 1 INTRODUCTION

Recent advancements in the reasoning capabilities of large language models have significantly accelerated progress in the field of Automated Theorem Proving (ATP) (Yang et al., 2024). Unlike traditional mathematical tasks, ATP requires models to start from a formalized theorem statement and construct rigorous logical proofs that can be verified within formal languages such as Lean (De Moura et al., 2015) and Isabelle (Paulson, 1994). However, the training of recent massive provers, such as DeepSeek-Prover (Ren et al., 2025) and Kimina-Prover (Wang et al., 2025), is hindered by the scarcity of formalized mathematical queries. Autoformalization addresses this by translating mathematical problems expressed in natural language into verifiable formal statements.

A significant challenge in autoformalization is the absence of a universal automatic evaluation standard (Li et al., 2024b). As widely recognized in previous work (Liu et al., 2025b), a valid formalization is supposed to meet two key criteria: (1) **Syntactic Validity**: the generated formal statements should be compiled successfully in the target formal language, and (2) **Semantic Consistency**: the translated statements should be semantically equivalent to the original mathematical problems. Earlier approaches primarily focus on autoformalization by directly prompting or utilizing in-context learning to instruct LLMs to generate valid formal statements (Azerbayev et al., 2023a; Yu et al., 2025). However, such approaches suffer from the limited formalization capabilities of LLMs, resulting in suboptimal results. Recent efforts shift towards training a specialized formalizer model from scratch (Wang et al., 2025; Wu et al., 2025; Lin et al., 2025). The formalizer is typically trained on extensive high-quality informal-formal pairs that are both syntactically valid and semantically consistent, demonstrating improved performance.

Despite the promise, existing formalization approaches tend to be suboptimal due to the following issues: (1) **Lack of Formal Knowledge**. The scarcity of formal language data in the pre-training corpora limits the foundational models' ability to inherently understand and generate formal statements effectively. Solely relying on post-training is insufficient for models to produce syntactically

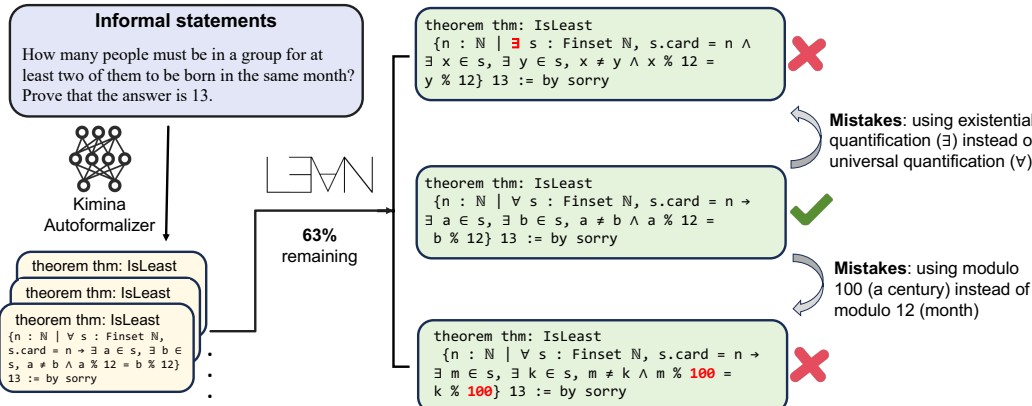

Figure 1: Illustration of Challenges in Autoformalization using Kimina-Autoformalizer. Approximately 40% of the statements fail to pass syntax validation, while the remaining statements tend to exhibit subtle misalignments with the original queries.

valid statements stably. For instance, Goedel-Formalizer-v2 (Lin et al., 2025), as the currently leading formalizer, achieves only 62.31% syntax pass@1 on combibench. Additionally, considering significant variations between different versions of formal languages (e.g. Lean 4 v.s. Lean 3), the trained formalizer often lacks generalizability across versions. (2) **Rough Consistency Validation**. Due to the high costs of manual data annotations, previous work relies on LLMs to assess the consistency between informal and formal expressions. However, the reliability of such LLMs-as-judge approach has not been thoroughly validated. As illustrated in fig. 1, generated formal statements exhibit subtle misalignments with their informal queries, which has also been observed in previous work (Wu et al., 2025), necessitating more precise and dependable consistency verification to ensure the semantic equivalence of translated statements.

To address these issues, we propose *Autoformalizer with Tool Feedback* (**ATF**) which integrates syntactic and consistency information as tools into the formalization process, thereby guiding models to adaptively refine the statements during generation. Specifically, we develop distinct tools for syntactic validity and semantic consistency. For syntactic validity, the tool processes formal statements and returns comprehensive compilation feedback from the Lean 4 compilers, offering precise guidance for syntax corrections. For semantic consistency, we benchmark the ability of LLMs to discern subtle misalignments between informal-formal pairs and implement a multi-LLMs-as-judge approach for consistency validation. The integration of syntactic information effectively compensates for the model's unfamiliarity with formal languages, allowing adjustments tailored to different language versions. Besides, the incorporation of consistency information helps the model to identify and address misalignments between informal and formal statements, enhancing semantic consistency. The training of ATF involves a cold-start phase on synthetic data to teach the model effective tool usage, an expert iteration phase to enhance the model's formalization capability and its ability to effectively implement revisions based on tool feedback, followed by a Direct Preference Optimization (DPO) phase to reduce ineffective revisions. Extensive experiments across three widely-used benchmarks demonstrate that ATF achieves substantial improvements over existing state-of-the-art formalizers (e.g., 29.13% semantic consistency improvement on CombiBench compared to the strongest Goedel-V2-Formalizer-32B). We further analyze the inference-time scaling properties of ATF and leverage it to synthesize formal statements from open-source mathematical queries, thereby contributing resources to advance future research in autoformalization and ATP.

The core components of this paper can be highlighted as:

- We develop two evaluation tools that effectively assess the validity of formal statements, providing accurate measurements of both syntactic validity and semantic consistency.

- We propose Autoformalizer with Tool Feedback (ATF), which enables models to invoke evaluation tools during the formalization process and adjust statements based on feedback, achieving superior results compared to existing baseline formalizers.

- We open-source Numina-ATF, a formal dataset containing 750K formal statements from Numina-v1.5 queries synthesized by ATF-32B (see Appendix D for details), supporting further development of formalizers and provers.

## 2    RELATED WORK

### 2.1    AUTOFORMALIZATION USING LLMS

Recent research in autoformalization has focused on using LLMs to accurately interpret and convert mathematical queries into structured formal languages, which can be categorized into two approaches: (1) Prompting powerful general models. Azerbayev et al. (2023a) and Wu et al. (2022) utilize In-context learning with carefully crafted formalization examples. Yu et al. (2025) directly asks DeepSeek-R1 (Guo et al., 2025) to transform mathematical queries into formal statements followed by data filtering to ensure the quality of the generated statements. This approach heavily relies on the inherent capabilities of the general model and the quality of prompts, resulting in poor performance. (2) Training a specialized model. Wang et al. (2025) leverage human-annotated cold-start data and employs an expert iteration strategy to train a lightweight Autoformalizer model. Lin et al. (2025) synthesize high-quality datasets containing reasoning paths from Claude 4, integrating reasoning capabilities into the autoformalization process. Wu et al. (2025) enhance the accuracy of the Autoformalizer by training on a combination of a knowledge-distilled dataset and a reasoning dataset. Though effective, this method requires extensive high-quality informal-formal pairs and faces challenges like data scarcity and limited adaptability across different formal languages. Our method adapts from (2) but innovatively incorporates formal standards directly into the generation process. By leveraging the model's reflective capabilities, it produces more accurate samples by refining statements using syntactic and consistency feedback.

### 2.2    TOOL-INTEGRATED RESONING

LLMs encounter limitations in tasks demanding precise calculation, faithful verification, or access to information beyond their knowledge. Tool-Integrated Reasoning (Lin & Xu, 2025) has emerged as a powerful approach to tackle these challenges, integrating external tools to enhance model performance. Proof assistants such as Lean 4 are well-suited to serve as tools that assist in ATP, which enables LLMs to access theorem databases and verify the correctness of the proof process. Li et al. (2024a) leverages retrieval-augmented generation (RAG) (Gao et al., 2023) to incorporate relevant theorems from formal libraries into the proof construction process, enhancing the precision and relevance of theorem proving. Ji et al. (2025) implement Lean 4-verifier into iterative refinement loops, allowing models to autonomously revise candidate proofs based on feedback, which complements the structural approach with continual improvement strategies. Different from the verification of proof, compiler results can not check the correctness of formalization. Our approach focuses on syntax and consistency to construct reliable tools that aid the formalization process.

## 3    METHODOLOGY

In this section, we systematically elaborate on the main components of ATF. We start with the construction of tools, detailing how we design reliable validity evaluation tools from the perspectives of syntax and consistency. Following this, we explain the ATF training pipeline, as depicted in fig. 2, which comprises three phases: an initial cold-start phase, an expert iteration phase, and a final Direct Preference Optimization (DPO) phase. During inference, ATF actively utilizes the developed tools to generate Lean 4 statements, iteratively modifying the output based on feedback until it passes both syntax and consistency checks.

### 3.1    TOOL DESIGN

To achieve reliable evaluation of formal statements, we design two distinct tools: **syntax check** for syntactic validity and **consistency check** for semantic consistency. The syntactic validity tool processes formal statements and returns detailed compilation feedback from Lean 4 compilers. We employ a pre-check stage and a grouped execution method to ensure more stable and rapid tool responses. The consistency check receives pairs of informal and formal statements and returns consistency results along with concise explanations. Notably, we implement a multi-LLMs-as-judge approach in consistency check, which is benchmarked to effectively discriminate minor inconsistencies in formal statements. Detailed implementation of tools can be found in Appendix A.

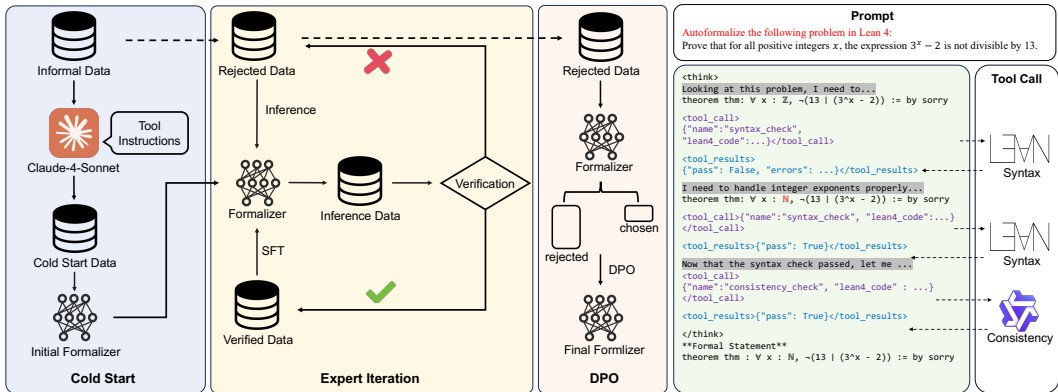

Figure 2: Framework of ATF consisting of three training stages: a cold-start phase to introduce basic tool usage with synthetic trajectories, followed by an expert iteration phase to refine formalization skills, and concluding with DPO to favor more effective paths with fewer revisions.

### 3.1.1 SYNTAX CHECK

Acquiring compiler feedback from Lean 4 (Moura & Ullrich, 2021) execution is an intuitive and direct method for syntactic validation checking, and it is widely used in ATP to verify the correctness of proofs. However, Lean 4 execution is quite time-consuming, struggling to handle large-scale statements. This presents a challenge in acquiring low-overhead and rapid execution responses.

To overcome such drawbacks in efficiency, we first implement a pre-check stage that filters out statements with obvious syntactic errors before compilation, such as missing necessary libraries and unmatched parentheses, aiming to reduce the workload. Additionally, we utilize a grouped method to enable batch execution of Lean 4 code. Statements are grouped based on import libraries. As fig. 3 shows, statements within the same group are concatenated into a single code file separated by namespaces. Then, Execution results are mapped to the corresponding statements based on their line numbers, en-

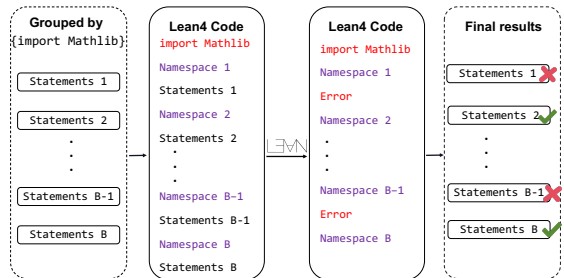

Figure 3: Grouped Lean 4 Execution.

abling efficient batch processing of Lean 4 code. In this paper, we adopt Lean 4 of version 4.15 [1], which is a stable version widely utilized in previous work (Yu et al., 2025).

### 3.1.2 CONSISTENCY CHECK

LLMs-as-judge methods have been widely adopted to replace manual evaluations for assessing the semantic consistency of formalizations (Huang et al., 2025). Although Wang et al. (2024) measure the abilities of different models to distinguish semantically correct formalizations, their capacities to discern subtle inconsistencies in similar formal statements remain underexplored.

To obtain more refined consistency feedback, we constructed a benchmark containing similar positive and negative statements. Specifically, we sampled 800 math queries from widely used formalization datasets (Azerbayev et al., 2023b; Zheng et al., 2021; Tsoukalas et al., 2024), where each query has a valid formal statement as the positive statement. We then instruct Gemini-2.5-Pro to generate perturbations for each statement, selecting 4 different perturbations as negative statements based on the following criteria:

- Character-level similarity with the positive statement is greater than 0.95.
- Syntactically valid but are not semantically consistent with the original statement.

---

[1]https://github.com/project-numina/kimina-lean-server

Table 1: Performance metrics of models in consistency checks, including Precision, Recall, FNR, TNR, and FPR.

| Model | Precision (↑) | Recall (TPR) (↑) | FNR (↓) | TNR (↑) | FPR (↓) |
|---|---|---|---|---|---|
| QWQ-32B | 0.8029 | 0.6758 | 0.3242 | 0.9171 | 0.0829 |
| Qwen3-32B | 0.7949 | **0.7367** | **0.2633** | 0.9050 | 0.0950 |
| Ensemble Vote | **0.8374** | 0.5967 | 0.4033 | **0.9421** | **0.0579** |

Following the setup in (Wang et al., 2024), we evaluate two widely used open-source models QWQ-32B and Qwen3-32B (Yang et al., 2025) in consistency checks on the benchmark. We find that an ensemble vote method (where consistency is only confirmed when both models give identical conclusions) effectively enhances the models' ability to discern subtle inconsistencies. As illustrated in table 1, while QWQ-32B and Qwen3-32B demonstrated comparable performance, their FPR was suboptimal (indicating that approximately 9% of inconsistent statements are misclassified). In contrast, the ensemble vote approach effectively reduces the FPR to below 6%. Therefore, we adopt this multi-LLMs-as-judge approach in this paper to ensure a more accurate consistency evaluation tool. More details about Benchmark constructions can be found in Appendix A.2.

## 3.2 TRAINING PIPELINE

With the tools constructed above, we implement the training pipeline for ATF using Qwen3-32B. The entire training process consists of three identical stages: a cold-start phase to teach the model the use of tools, an expert iteration phase to enhance the model's formalization capabilities, and a Direct Preference Optimization (DPO) phase to reduce ineffective revisions found in multiple iterations.

**Data** The whole training process of ATF is conducted based on the NuminaMath-1.5 dataset (LI et al., 2024), which contains approximately 900k competition-level math problems ranging from Chinese high school math exercises to international mathematics Olympiad competition problems. Following (Wang et al., 2025), we select a challenging subset of NuminaMath-1.5, consisting of 178k entries that cover several competition-level data sources. Detailed information about different data sources can be found in table 2.

Table 2: Overview of data sources.

| Source | Size |
|---|---|
| amc_aime | 1,805 |
| olympiads_ref | 3,221 |
| cn_contest | 20,791 |
| olympiad | 152,652 |
| Total | 178,469 |

**Cold Start for Tool Integration** The cold-start phase begins by prompting Claude-4-Sonnet to generate multi-turn tool invocation reasoning paths. Considering that the consistency check using multi-LLMs-as-judge is more time-consuming compared to the syntax check, we establish several rules for more efficient revisions:

- Consistency check is only permitted after syntax check passes.
- Syntax check must be invoked first after revision.
- The process stops if and only if both the syntax and consistency checks pass.

Following the default format of Claude, the tool invocations and returned information are respectively enclosed in <tool_call></tool_call> and <tool_result></tool_result> tags. To enable multiple rounds of tool calls and revisions within a limited context length, we prompt Claude to perform concise reasoning. Ultimately, we extract approximately 10% queries from the dataset for data synthesis, and upsample the reasoning paths with more than one revision to obtain a final 24K cold-start dataset. We choose Qwen3-32B as the foundation model and mask the losses on tool result tokens in training to prevent the model from directly mimicking the tool executions.

**Expert Iteration for Formalization Capability Enhancement** After fine-tuning the formalizer on the synthetic cold-start dataset, the model is familiar with the format of tool invocation and has acquired the behavior of making revisions based on error feedback from tools. Next, we conduct an expert iteration training on the remaining data, aiming to further improve the model's formalization

capabilities. Following a standard expert iteration pipeline, in each iteration, we use the current model to generate formalization attempts on the remaining math queries and filter out those that violate the tool invocation rules above. We collect all successful formalization trajectories with the revision attempts $< 8$ and merge them with data from previous rounds for training, retaining failed queries for the next iteration. We conduct training from the base model in each iteration, and the configuration remains the same as in the cold-start phase.

**DPO for Effective Revision** After expert iteration, the formalizer excels in generating valid statements through iterative tool calling and revision. However, we observe that the model sometimes exhibits consecutive identical errors (e.g., the same syntax error appearing multiple times without being resolved). To guide the model to reduce such ineffective revisions, we further conduct a DPO (Rafailov et al., 2023) training to encourage the model to complete formalization in fewer attempts. Specifically, we first perform self-sampling on the remaining data from the expert iteration. For each math query, we select the trajectory with fewer revision attempts as the positive sample and the one with more revision attempts as the negative sample (maintaining their revision attempt difference $\geq$ 3), finally collecting 10K pairs. In addition to masking the loss on tool result tokens, we also mask tool invocation-related tokens to prevent instability in tool invocation behavior. Considering that the pairs share similar distributions, we adopt the DPO loss along with a negative log-likelihood (NLL) loss (Dubey et al., 2024) on chosen trajectories to avoid the decline of chosen reward in training

$$\mathcal{L} = -\mathbb{E}\left[\log \sigma \left(\beta \log \frac{\pi_\theta(y_w|x)}{\pi_{\text{ref}}(y_w|x)} - \beta \log \frac{\pi_\theta(y_l|x)}{\pi_{\text{ref}}(y_l|x)}\right)\right] - \alpha \mathbb{E}\left[\log \pi_\theta(y_w|x)\right], \quad (1)$$

where $\pi_\theta$ is the policy model, $\pi_{\text{ref}}$ is the reference model, $y_w$ and $y_l$ are the chosen and rejected trajectories respectively, $\beta$ is the temperature parameter, $\alpha$ is the weighting coefficient, and $\sigma$ represents the sigmoid function. Notably, we adopt DPO instead of online reinforcement learning algorithms such as GRPO (Shao et al., 2024), because the model has already achieved strong capabilities after expert iteration, resulting in a relatively low proportion of negative trajectories in self-sampling, which makes DPO more efficient.

## 4 EXPERIMENTS

### 4.1 EXPERIMENTAL SETUP

**Training Configurations** For the SFT training involved in cold-start and expert iteration phases, we conduct full parameter fine-tuning on 128 NPUs with an initial learning rate of 2e-5, using a cosine decay style that decays to 1e-7 over 3 epochs. The DPO training is conducted with hyper-parameters ($\beta = 0.1$) and ($\alpha = 0.3$) on 128 NPUs for 1 epoch. In addition to the ATF-32B model, we also train an ATF-8B-Distilled using the same data. Detailed training parameters are listed in Appendix B.

**Baselines** For baseline comparisons, we evaluate ATF against a series of the most performant formalizer models, including: Kimina-Autoformalizer-7B (Wang et al., 2025), StepFun-Formalizer-7B/32B (Wu et al., 2025), and Goedel-V2-Formalizer-8B/32B (Lin et al., 2025).

**Evaluations** Following Wu et al. (2025), we select three widely-used ATP datasets for evaluation, including two in-distribution datasets: FormalMath-Lite (Yu et al., 2025) and ProverBench (Ren et al., 2025), along with an out-of-distribution dataset CombiBench (Liu et al., 2025a). To ensure fair comparison, we perform similarity-based decontamination on all training data against these evaluation sets. In terms of evaluation metrics, we assess both syntactic validity and consistency validity of generated statements using the tools designed above (only syntactically valid statements proceed to consistency evaluation). For each math query, we sample 16 times with temperature = 0.6 and report unbiased Pass@1, Pass@8, and Pass@16 pass rates. For ATF we set the max revision attempts $< 4$ which results in output lengths roughly equivalent to those of Goedel-V2-Formalizer-32B. Considering the limitations of LLMs-as-judge in terms of consistency check, we further conduct human evaluation on 32B-scale models as the gold standard. Specifically, we randomly sample 100 instances from each benchmark, and each instance is evaluated by 3 experts independently, with the majority opinion serving as the final judgment. Detailed evaluation procedures are provided in Appendix C.

Table 3: Performance comparison of different Formalizers. SC represents Syntax Check pass rate (%) and CC represents Consistency Check pass rate (%). The best results are presented in **bold** and the second with underline.

| Model | FormalMath-Lite | | ProverBench | | CombiBench | |
|---|---|---|---|---|---|---|
| | SC | CC | SC | CC | SC | CC |
| **Pass@1** | | | | | | |
| Kimina-Autoformalizer-7B (Wang et al., 2025) | 93.11 | 64.77 | 86.44 | 45.95 | 63.19 | 14.00 |
| StepFun-Formalizer-7B (Wu et al., 2025) | 89.31 | 69.72 | 71.52 | 45.16 | 46.94 | 16.50 |
| Goedel-V2-Formalizer-8B (Lin et al., 2025) | 95.25 | 83.06 | 92.36 | 78.64 | 59.94 | 31.44 |
| StepFun-Formalizer-32B (Wu et al., 2025) | 91.87 | 73.11 | 69.73 | 49.70 | 55.19 | 25.50 |
| Goedel-V2-Formalizer-32B (Lin et al., 2025) | 95.72 | 85.41 | 91.39 | 79.70 | 62.31 | 36.25 |
| ATF-8B-Distilled | 96.55 | 91.12 | 91.49 | 85.16 | 74.00 | 51.69 |
| ATF-32B | **97.94** | **94.51** | **94.29** | **89.78** | **86.69** | **65.38** |
| **Pass@8** | | | | | | |
| Kimina-Autoformalizer-7B (Wang et al., 2025) | 99.42 | 86.81 | 95.82 | 65.51 | 93.72 | 26.56 |
| StepFun-Formalizer-7B (Wu et al., 2025) | 96.82 | 88.22 | 86.46 | 68.59 | 78.11 | 36.17 |
| Goedel-V2-Formalizer-8B (Lin et al., 2025) | 98.67 | 96.23 | 96.28 | 94.03 | 90.77 | 57.78 |
| StepFun-Formalizer-32B (Wu et al., 2025) | 98.05 | 90.63 | 89.03 | 74.71 | 83.14 | 46.10 |
| Goedel-V2-Formalizer-32B (Lin et al., 2025) | 99.21 | 97.68 | 96.28 | 94.27 | 90.64 | 70.58 |
| ATF-8B-Distilled | 99.95 | 98.82 | 99.11 | 97.24 | 97.49 | 81.88 |
| ATF-32B | **99.97** | **99.27** | **99.66** | **98.20** | **98.73** | **92.22** |
| **Pass@16** | | | | | | |
| Kimina-Autoformalizer-7B (Wang et al., 2025) | 99.76 | 90.67 | 96.52 | 69.13 | 97.00 | 30.00 |
| StepFun-Formalizer-7B (Wu et al., 2025) | 97.37 | 90.67 | 89.13 | 73.91 | 84.00 | 44.00 |
| Goedel-V2-Formalizer-8B (Lin et al., 2025) | 99.04 | 97.13 | 96.52 | 94.78 | 96.00 | 66.00 |
| StepFun-Formalizer-32B (Wu et al., 2025) | 98.80 | 93.54 | 91.30 | 79.57 | 87.00 | 51.00 |
| Goedel-V2-Formalizer-32B (Lin et al., 2025) | 99.52 | 98.80 | 96.52 | 95.22 | 93.00 | 79.00 |
| ATF-8B-Distilled | **100.00** | **99.52** | 99.57 | 97.83 | 99.00 | 87.00 |
| ATF-32B | **100.00** | **99.52** | **100.00** | **98.70** | **100.00** | **96.00** |
| **Human Evaluation** | | | | | | |
| Kimina-Autoformalizer-7B (Wang et al., 2025) | 92.00 | 70.00 | 87.00 | 59.00 | 59.00 | 7.00 |
| StepFun-Formalizer-32B (Wu et al., 2025) | 92.00 | 75.00 | 70.00 | 57.00 | 53.00 | 18.00 |
| Goedel-V2-Formalizer-32B (Lin et al., 2025) | 97.00 | 92.00 | 93.00 | 81.00 | 60.00 | 22.00 |
| ATF-32B | **98.00** | **95.00** | **94.00** | **85.00** | **90.00** | **49.00** |

## 4.2 MAIN RESULTS

Based on the main experimental results listed in table 3, we draw several summarizations:

**ATF consistently outperforms all baseline models across all benchmarks on both syntax and consistency metrics.** ATF-32B achieves superior performance on all three evaluation datasets, with particularly notable improvements in semantic consistency. For example, ATF-32B achieves Pass@1 consistency scores of 94.51% on FormalMath-Lite, 89.78% on ProverBench, and 65.38% on CombiBench, consistently surpassing the best baseline Goedel-V2-Formalizer-32B by margins of 9.1%, 10.08%, and 29.13% respectively.

**ATF demonstrates strong generalization capabilities in out-of-distribution scenarios.** The substantial performance improvements are particularly evident on CombiBench, which contains diverse combinatorial mathematics problems that significantly differ from the training data distribution. While most baseline models struggle significantly on this dataset (e.g., StepFun-Formalizer-32B achieves only 25.50% Pass@1 in consistency), ATF-32B maintains robust performance at 65.38%, indicating strong generalization beyond the training distribution.

**ATF benefits significantly from increased sampling (Pass@k) and maintains excellent performance even at the 8B scale.** The performance gains become more pronounced at higher sampling rates, with ATF-32B achieving remarkable scores (>96%) on consistency checks across all benchmarks at Pass@16. Moreover, ATF-8B-Distilled demonstrates remarkable efficiency, achieving 91.12% Pass@1 consistency on FormalMath-Lite while using significantly fewer parameters than 32B baseline models, showcasing the effectiveness of our training methodology.

Table 4: Ablation study of ATF components across different benchmarks using the pass@1 (%) metric.

| Components | FormalMath-Lite | | ProverBench | | CombiBench | |
|---|---|---|---|---|---|---|
| | SC | CC | SC | CC | SC | CC |
| **SYNTAX CHECK + CONSISTENCY CHECK** | | | | | | |
| *Cold Start* | 93.91 | 89.01 | 91.25 | 84.32 | 61.50 | 42.44 |
| *+ Expert Iteration* | 97.85 | 94.15 | 94.16 | 89.10 | 85.12 | 63.88 |
| *+ DPO* | 97.94 | 94.51 | 94.29 | 89.78 | 86.69 | 65.38 |
| **SYNTAX CHECK ONLY** | | | | | | |
| *Cold Start* | 93.91 | 73.77 | 91.25 | 67.96 | 61.50 | 27.12 |
| *+ Expert Iteration* | 97.85 | 80.26 | 94.16 | 75.54 | 85.12 | 40.12 |
| *+ DPO* | 97.94 | 81.13 | 94.29 | 75.68 | 86.69 | 41.68 |
| **NO TOOLS** | | | | | | |
| *Cold Start* | 79.22 | 64.23 | 66.39 | 51.25 | 35.00 | 16.06 |
| *+ Expert Iteration* | 86.39 | 71.96 | 73.75 | 60.54 | 48.38 | 22.88 |
| *+ DPO* | 86.77 | 72.89 | 73.89 | 60.92 | 50.81 | 23.69 |

**Human evaluation validates both the effectiveness of the consistency check tool and the superior performance of ATF.** Although the strictness of the multi-LLMs-as-judge method results in some sacrifices in recall (see table 1), ATFstill consistently outperforms the baselines, especially on CombiBench where existing models achieve a maximum pass rate of only 22%. We further compute the Pearson correlation coefficient (Sedgwick, 2012) between the results obtained from the consistency check tool and those from human evaluation, yielding a coefficient of 0.746. This value indicates a strong positive linear relationship, confirming the reliability of our consistency check.

### 4.3 ABLATIONS

To understand the contribution of each component in ATF, we conduct comprehensive ablation studies by systematically removing different elements. Table 4 presents the results across three configurations: full ATF with both syntax and consistency checks, syntax check only, and no tools.

**Tool feedback is essential for effective formalization.** The comparison between different tool configurations demonstrates the critical importance of our designed tools. Without any tool guidance, performance drops dramatically across all benchmarks. For example, on CombiBench consistency check, the no-tools configuration achieves only 23.69% Pass@1 compared to 65.38% with full tool feedback. Adding the consistency check on top of the syntax check provides further substantial gains, improving ProverBench consistency from 75.68% to 89.78%, highlighting that semantic validation is crucial beyond syntactic correctness.

**Progressive training stages yield cumulative improvements.** Each training phase contributes meaningfully to the final performance. Expert iteration provides the most substantial improvements over cold start (e.g., CombiBench consistency improves from 42.44% to 63.88%), while DPO further offers additional refinements. This staged approach effectively teaches the model tool usage, enhances formalization capabilities, and improves efficiency.

## 5 ANALYSIS

### 5.1 SCALING ANALYSIS

While our main experiments report performance under limited resource constraints, we further investigate ATF's inference time scaling behavior by extending both the revision attempts and the parallel sampling counts K. Since syntax check already achieves high pass rates, we focus our analysis on consistency check performance. As illustrated in fig. 4a ,although ATF is trained with revision attempts limited to fewer than 8, performance continues to improve gradually as the number of revision attempts increases. This suggests that the model has learned effective revision strategies that

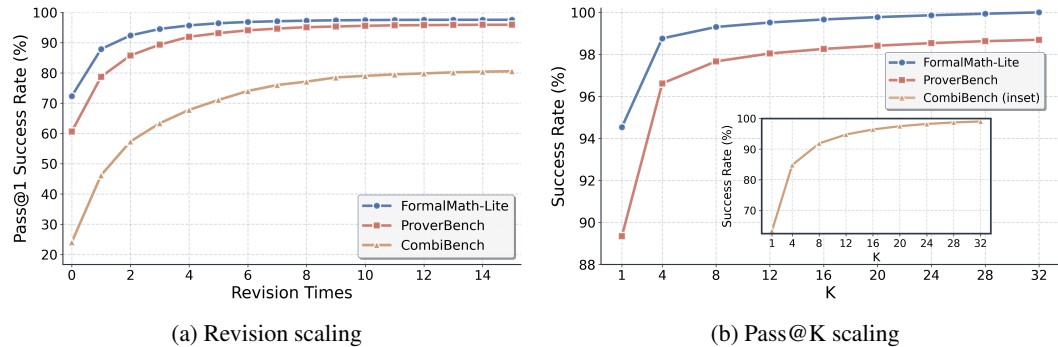

(a) Revision scaling

(b) Pass@K scaling

Figure 4: Inference time scaling of ATF.

generalize beyond the training constraints, enabling it to iteratively refine statements toward higher semantic consistency. Additionally, ATF can further benefit from increased parallel sampling (from Pass@1 to Pass@32 in fig. 4b), achieving 100% pass rates on CombiBench. This scaling behavior indicates that our tool-integrated approach not only improves individual formalization quality but also enhances the diversity and coverage of valid formalizations across multiple attempts.

## 5.2 TOOL ANALYSIS

We continue to analyze behavior of ATF when invoking the two types of tools. As shown in fig. 5, tool usage varies significantly across datasets, with CombiBench requiring the highest average tool calls (8.35) due to its combinatorial complexity, while FormalMath-Lite requires fewer attempts (3.19), demonstrating ATF's ability to adapt revision intensity based on problem difficulty. The consistency check generally exhibits lower pass rates compared to the syntax check, reflecting the inherent difficulty of semantic alignment versus rule-based syntactic correctness. However, Prover-Bench presents an exception where consistency check (66.34%) outperforms syntax check (61.65%), primarily due to the dataset's extensive calculus-related queries that introduce additional syntactic complexity in formal representations. We also find that the consistency check success rate consistently decreases with increasing revision attempts across all datasets from 69.5% on the first attempt to 8.8% on the 8th attempt. This declining pattern suggests that while ATF can identify semantic inconsistencies, the revision process becomes increasingly challenging as the model exhausts its most confident revision strategies.

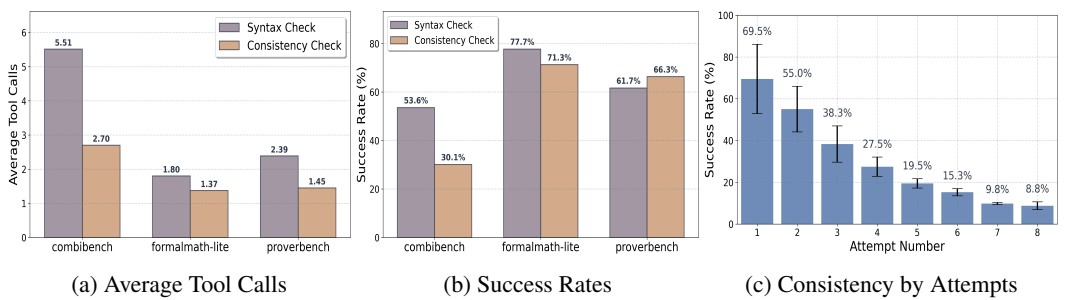

(a) Average Tool Calls

(b) Success Rates

(c) Consistency by Attempts

Figure 5: Comparative Analysis of Tool Usage Metrics

## 6 CONCLUSION

In this paper, we presented *Autoformalizer with Tool Feedback* (ATF), a novel framework that enhances autoformalization by incorporating syntactic and semantic validation tools into the generation process to compensate for insufficient formal knowledge and unreliable semantic validation in previous formalizers. Experiments across three benchmarks show that ATFsubstantially outperforms existing formalizers and demonstrates promising scaling behavior during inference. We contribute an open-source dataset of 750K formal statements derived from competition-level mathematical queries, facilitating future research in autoformalization and automated theorem proving.

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

## A   DETAILS OF THE TOOL IMPLEMENTATION

### A.1   SYNTAX CHECK

Specifically, the pre-check stage deployed in syntax check aims to filter out the illegal formalization before the Lean 4 Execution, including:

- Missing necessary libraries explicitly indicated in the prompt.
- Invalid statements that do not end with "by sorry".
- Statements with unmatched parentheses.

Table 5: Comparison of Average Execution Time per Statement.

| Method | Time (s) |
|---|---|
| individual | 6.215 |
| grouped | 0.808 |

Although the proportion of statements filtered by the pre-check is only approximately 2%, it helps the subsequent grouped Lean 4 execution. This is because we find that the missing "by sorry" and unmatched parentheses will lead to execution timeouts. During the grouped Lean 4 execution stage, we set the batch size to 20 and a timeout of 300s for each execution. For batches that encounter exceptions, we conduct individual calls for each statement in the batch to ensure the accuracy of the syntax check. We conduct a fair comparison of the efficiency between grouped Lean 4 execution and individual statement execution when processing 100K statements on a machine with 8 CPUs and 100GB of memory. The results indicate that the average processing time per statement is reduced by 87% (from 6.2s to 0.8s in table 5)

### A.2   CONSISTENCY CHECK

The construction of the benchmark for the consistency check involves the following datasets:

- FormalMATH-All (Yu et al., 2025): A comprehensive dataset consisting of a wide range of formal mathematical statements synthesized by LLMs.
- MiniF2F (Zheng et al., 2021): A dataset created for evaluating automated theorem proving techniques, featuring a collection of mathematical problems designed for human challenge.
- PutnamBench (Tsoukalas et al., 2024): A dataset derived from the prestigious Putnam Mathematical Competition.

Each instance contains an informal math query and a formal statement. Initially, we filter out instances that fail in Lean 4 execution. Then we employ GPT-4o to categorize each statement into specific domains, including Algebra, Calculus, Combinatorics, Geometry, Inequalities, Logic and Puzzles, Number Theory, and Other. Subsequently, 100 data instances are randomly selected from each domain to serve as seeds for perturbation. We manually devise different types of perturbations (Appendix G.1) and instruct Gemini-2.5-Pro to apply equivalent perturbations to each statement. To enhance the difficulty of the benchmark, we reject perturbations that share a Levenshtein similarity (Zhang et al., 2017) with the original statement greater than 0.95. We also make sure that the perturbations are not equivalent to the reference statement through human verification. Finally, we select 100 instances along with four different random perturbations for each domain, collecting 800 instances in total.

In implementations, we require the LLMs to provide brief explanations while assessing equivalence to aid model understanding, using the same prompt as Wang et al. (2024) (Appendix G.2). When both QWQ-32B and Qwen3-32B provide a judgment of inconsistency, the response from QWQ-32B is prioritized as the result of the consistency check. In cases where the two models differ, the response indicating inconsistency is used as the results.

## B   HYPERPARAMETER OF TRAINING

The specific hyperparameter settings for ATF training are listed in table 6.

Table 6: Training Hyperparameters for ATF-32B

(a) Cold Start & Expert Iteration

| Hyperparameter | Value |
|---|---|
| Epochs | 3 |
| Batch Size | 128 |
| Max Length | 40,960 |
| Learning Rate (LR) | $2 \times 10^{-5}$ |
| Minimum Learning Rate | $1 \times 10^{-7}$ |
| LR Decay Style | Cosine |
| LR Warmup Fraction | 0.1 |
| Adam Beta1 | 0.9 |
| Adam Beta2 | 0.95 |
| Weight Decay | $1 \times 10^{-2}$ |
| Gradient Clipping | 1.0 |

(b) DPO

| Hyperparameter | Value |
|---|---|
| Epochs | 1 |
| Batch Size | 64 |
| Max Length | 40,960 |
| Learning Rate (LR) | $2 \times 10^{-6}$ |
| Minimum Learning Rate | $1 \times 10^{-7}$ |
| LR Decay Style | Cosine |
| LR Warmup Fraction | 0.03 |
| Adam Beta1 | 0.9 |
| Adam Beta2 | 0.99 |
| Weight Decay | $1 \times 10^{-2}$ |
| Gradient Clipping | 1.0 |
| Temperature Weighting | 0.1 |
| NLL Loss Weighting | 0.3 |

## C  EVALUATION DETAILS

The benchmarks used in the main experiments are listed:

- FormalMath-Lite (Yu et al., 2025): a streamlined version of the FormalMath benchmark, focusing on essential formal mathematical statements.

- Proverbench (Ren et al., 2025): a challenging datasets proposed by DeepSeek-Prover-V2, containing a diverse set of logical statements.

- Combibench (Liu et al., 2025a): A benchmark tailored for combinatorial problem solving.

We select these three benchmarks for their timeliness, which minimizes the likelihood of data contamination with existing baselines.

Since these benchmarks are designed for ATP, there are multiple data instances having the same mathematical query, or a single query containing multiple subproblems. To address this, we manually perform the following operations:

- For duplicate queries, we retain only the first one.

- For queries with multiple subproblems, we manually split them into several queries.

Table 7 shows the amount of data before and after processing for each benchmark.

Table 7: Data Amount Before and After Processing for Each Benchmark

| Benchmark | Before Processing | After Processing |
|---|---|---|
| FormalMath-Lite | 425 | 418 |
| Proverbench | 325 | 230 |
| Combibench | 100 | 100 |

To prevent data contamination, we employ the all-MiniLM-L6-v2 model to compute embedding similarities between our training data and the benchmark above. We subsequently exclude training samples that exhibit a cosine similarity $> 0.8$ with the benchmark data. In terms of Human Evaluations, we invite three experts whose research areas involve formal languages. Each person is provided with the original mathematical query and the formal statement, as well as the results from the consistency check execution. They are allowed to refer to the explanation presented in the tool invocation results or ask Claude 4 for additional help. Each participant has been paid according to the number of queries they handled.

## D DETAILS OF NUMINA-ATF

Numina-V1.5 (LI et al., 2024) contains a substantial collection of high-quality competition-level mathematical problems and has been extensively utilized as training data in ATP. However, the original dataset only includes mathematical queries expressed in natural language, lacking corresponding formalized statements. Although some open-source formal statement datasets exist, they either suffer from low quality `https://huggingface.co/datasets/Goedel-LM/Goedel-Pset-v1` or cover only a limited subset `https://huggingface.co/datasets/AI-MO/NuminaMath-LEAN`. To facilitate subsequent related research, we employed ATF-32B to perform statement synthesis on the complete Numina-V1.5 dataset, yielding 752,553 informal-formal pairs that have been validated for both syntax and consistency.

## E THE USE OF LARGE LANGUAGE MODELS

Large language models are used solely as writing assistance tools to improve sentence clarity, format tables, and enhance readability. LLMs do not contribute to the core research contributions, experimental design, or scientific content of this work. The authors take full responsibility for all content presented in this paper.

## F CASE STUDY

---

**Case Study: Ladies' Club Partition into Two Cliques**

**Natural Language Problem:**
There are 100 members in a ladies' club. Each lady has had tea (in private) with exactly 56 of the other members of the club. The Board, consisting of the 50 most distinguished ladies, has all had tea with one another. Prove that the entire club may be split into two groups in such a way that, within each group, any lady has had tea with any other.

**ATF-32B Output:**
<think>
Looking at this problem, I need to formalize a graph theory statement about partitioning a graph into two cliques. Let me analyze the key components:
1. 100 ladies in a club (vertices in a graph)
2. Each lady has had tea with exactly 56 others (each vertex has degree 56)
3. The 50 most distinguished ladies all had tea with each other (a clique of size 50)
4. We need to prove the club can be split into two groups, where each group forms a clique
This is equivalent to showing the complement graph is bipartite.
Let me write the initial Lean 4 statement:

```lean
import Mathlib
import Aesop

theorem my_favorite_theorem :
  ∀ (G : SimpleGraph (Fin 100)) [DecidableRel G.Adj],
  (∀ v : Fin 100, G.degree v = 56) →
  (∃ (S : Finset (Fin 100)), S.card = 50 ∧ ∀ u v ∈ S, G.Adj u v) →
  ∃ (A B : Finset (Fin 100)), A ∪ B = Finset.univ ∧ A ∩ B = ∅ ∧
    (∀ u v ∈ A, G.Adj u v) ∧ (∀ u v ∈ B, G.Adj u v) := by sorry
```

```
<tool_calls>
{
  "name": "syntax_check",
  "arguments": {
  "lean4_code": "import Mathlib\nimport Aesop\n\ntheorem my_favorite_theorem : \n
    ∀ (G : SimpleGraph (Fin 100)) [DecidableRel G.Adj],\n  (∀ v : Fin 100, G.
  degree v = 56) →\n  (∃ (S : Finset (Fin 100)), S.card = 50 ∧ ∀ u v ∈ S, G.Adj u
  v) →\n  ∃ (A B : Finset (Fin 100)), A ∪ B = Finset.univ ∧ A ∩ B = ∅ ∧ \n    (∀
  u v ∈ A, G.Adj u v) ∧ (∀ u v ∈ B, G.Adj u v) := by sorry"}
}
</tool_calls>

<tool_results>
Function: syntax_check
```

---

```
Output: {
  "pass": false,
  "errors": [
    {
      "severity": "error",
      "pos": {
        "line": 7,
        "column": 48
      },
      "endPos": {
        "line": 7,
        "column": 50
      },
      "data": "unexpected token '∈'; expected ','"
    }
  ]
}
</tool_results>
```

The syntax check failed due to incorrect quantifier syntax. I need to fix the '∀ u v ∈ A' and '∀ u v ∈ B' expressions. Let me correct this:

```
import Mathlib
import Aesop

theorem my_favorite_theorem :
  ∀ (G : SimpleGraph (Fin 100)) [DecidableRel G.Adj],
  (∀ v : Fin 100, G.degree v = 56) →
  (∃ (S : Finset (Fin 100)), S.card = 50 ∧ ∀ u ∈ S, ∀ v ∈ S, G.Adj u
v) →
  ∃ (A B : Finset (Fin 100)), A ∪ B = Finset.univ ∧ A ∩ B = ∅ ∧
    (∀ u ∈ A, ∀ v ∈ A, G.Adj u v) ∧ (∀ u ∈ B, ∀ v ∈ B, G.Adj u v) :=
by sorry
```

```
<tool_calls>
{
  "name": "syntax_check",
  "arguments": {
  "lean4_code": "import Mathlib\nimport Aesop\n\ntheorem
my_favorite_theorem : \n  ∀ (G : SimpleGraph (Fin 100)) [DecidableRel
G.Adj],\n  (∀ v : Fin 100, G.degree v = 56) →\n  (∃ (S : Finset (Fin
100)), S.card = 50 ∧ ∀ u ∈ S, ∀ v ∈ S, G.Adj u v) →\n  ∃ (A B : Finset
(Fin 100)), A ∪ B = Finset.univ ∧ A ∩ B = ∅ ∧ \n    (∀ u ∈ A, ∀ v ∈ A,
G.Adj u v) ∧ (∀ u ∈ B, ∀ v ∈ B, G.Adj u v) := by sorry"}
}
</tool_calls>
```

```
<tool_results>
Function: syntax_check
Output: {
  "pass": true,
  "errors": []
}
</tool_results>
```

Now let me verify the consistency with the corrected code:

```
<tool_calls>
{
  "name": "consistency_check",
  "arguments": {
  "lean4_code": "import Mathlib\nimport Aesop\n\ntheorem
my_favorite_theorem : \n  ∀ (G : SimpleGraph (Fin 100)) [DecidableRel
G.Adj],\n  (∀ v : Fin 100, G.degree v = 56) →\n  (∃ (S : Finset (Fin
100)), S.card = 50 ∧ ∀ u ∈ S, ∀ v ∈ S, G.Adj u v) →\n  ∃ (A B : Finset
```

```
(Fin 100)), A ∪ B = Finset.univ ∧ A ∩ B = ∅ ∧ \n      (∀ u ∈ A, ∀ v ∈ A,
G.Adj u v) ∧ (∀ u ∈ B, ∀ v ∈ B, G.Adj u v) := by sorry"}
}
</tool_calls>

<tool_results>
Function: consistency_check
Output: {
  "pass": true,
  "explanations": "1. Mathematical Text Analysis: The problem involves
a graph of 100 nodes (club members) where each node has degree 56.
There exists a subset S of 50 nodes (the Board) forming a clique. The
goal is to partition the graph into two cliques A and B. Key elements:
100 members, degree 56 for all, a 50-node clique subset, and
partitioning into two cliques.\n2. Lean4 Code Analysis: The theorem
formalizes a SimpleGraph on Fin 100 (100 nodes). The premises require ∀
v, degree v = 56, and ∃S (Finset) with |S|=50 and ∀u,v ∈ S, Adj u v.
The conclusion requires ∃A,B (Finsets) partitioning Finset.univ (A ∪ B =
 univ, A ∩ B = ∅) where ∀u,v ∈ A, Adj u v and similarly for B.\n3.
Comparative Analysis: All structural elements match. The 100 members
are modeled via Fin 100. The degree condition is correctly captured.
The Board's clique is formalized via ∃S with card 50 and mutual
adjacency. The partition into two cliques is accurately represented via
A and B with the required properties. No semantic drift or missing
elements detected. Lean syntax is valid and precise.\n4. Accuracy
Confirmation: The formalization is exact. All components of the problem
(node count, degree, Board clique, partition into two cliques) are
preserved without additions or omissions."
}
</tool_results>
```

</think>
**Final Statement**

```
import Mathlib
import Aesop

theorem my_favorite_theorem :
  ∀ (G : SimpleGraph (Fin 100)) [DecidableRel G.Adj],
  (∀ v : Fin 100, G.degree v = 56) →
  (∃ (S : Finset (Fin 100)), S.card = 50 ∧ ∀ u ∈ S, ∀ v ∈ S, G.Adj u
v) →
  ∃ (A B : Finset (Fin 100)), A ∪ B = Finset.univ ∧ A ∩ B = ∅ ∧
    (∀ u ∈ A, ∀ v ∈ A, G.Adj u v) ∧ (∀ u ∈ B, ∀ v ∈ B, G.Adj u v) :=
by sorry
```

**Goedel-V2-Formalizer-32B:**

```
import Mathlib
import Aesop

set_option maxHeartbeats 0

open BigOperators Real Nat Topology Rat

theorem my_favorite_theorem :
  ∀ (ladies : Finset ℕ) (tea_relation : ℕ → ℕ → Prop),
  ladies.card = 100 →
  (∀ l ∈ ladies, (ladies.filter (λ x => tea_relation l x)).card = 56) →
  (∀ a b, a ∈ ladies → b ∈ ladies → tea_relation a b → tea_relation
b a) →
  ∃ (board : Finset ℕ), board ⊆ ladies ∧ board.card = 50 ∧
    (∀ a b, a ∈ board → b ∈ board → a ≠ b → tea_relation a b) →
  ∃ (group1 group2 : Finset ℕ),
    group1 ∪ group2 = ladies ∧
```

```
      group1 ∩ group2 = ∅ ∧
      (∀ a b, a ∈ group1 → b ∈ group1 → a ≠ b → tea_relation a b) ∧
      (∀ a b, a ∈ group2 → b ∈ group2 → a ≠ b → tea_relation a b) :=
by sorry
```

(**Inconsistent.** The formalization's logical structure is incorrect. It presents the existence of the board as a conditional derivation rather than as a given precondition. This changes the logical structure of the problem, making the formalization incorrect.)

**Kimina-Autoformalizer-7B:**

```
import Mathlib

theorem my_favorite_theorem
    (tea : Fin 100 → Fin 100 → Prop)
    (h_sym : ∀ i j, tea i j = tea j i)
    (h_antisymm : ∀ i j, i ≠ j → tea i j ∨ tea j i)
    (h_antisymm' : ∀ i j, i ≠ j → (tea i j → tea j i))
    (h_antisymm'' : ∀ i j, i ≠ j → (tea j i → tea i j))
    (h_antisymm''' : ∀ i j, i ≠ j → (tea i j → ¬tea j i))
    (h_antisymm'''' : ∀ i j, i ≠ j → (¬tea i j → tea j i)) :
    ∃ s1 s2 : Finset (Fin 100), s1 ∪ s2 = Finset.univ ∧
    s1 ∩ s2 = ∅ ∧ ∀ i ∈ s1, ∀ j ∈ s1, i ≠ j → tea i j ∧
    ∀ i ∈ s2, ∀ j ∈ s2, i ≠ j → tea i j := by sorry
```

(**Inconsistent.** This formalization lacks two core conditions of the problem (degree 56 and the existence of a 50-person complete subgraph) and contains contradictory assumptions.)

**StepFun-Formalizer-32B:**

```
import Mathlib
import Aesop

theorem my_favorite_theorem
    (tea : Fin 100 → Fin 100 → Prop)
    (h1 : ∀ i j, tea i j = tea j i)
    (h2 : ∀ i, ∃! j, tea i j)
    (h3 : ∃ b : Finset (Fin 100), b.card = 50 ∧ ∀ i ∈ b, ∀ j ∈ b, tea
i j) :
    ∃ s1 s2 : Finset (Fin 100), s1 ∪ s2 = Finset.univ ∧
    s1 ∩ s2 = ∅ ∧
    ∀ i ∈ s1, ∀ j ∈ s1, tea i j ∧
    ∀ i ∈ s2, ∀ j ∈ s2, tea i j := by sorry
```

(**Inconsistent.** "Each person drinks tea with 56 people" is incorrectly formalized as "Each person drinks tea with 1 person.")

# G  PROMPTS

## G.1  PROMPT FOR STATEMENT PERTURBATIONS

---

**Prompt for Statement Perturbations**

Role: Lean4 Semantic Perturbation Expert

Input:

- Original_Lean4Code: A Lean 4 theorem statement that needs to be perturbed.

Goal: Generate 6 different semantic perturbations of the given Lean4 statement. Each perturbation should:

- Change the semantic meaning while maintaining syntactic similarity
- Be syntactically valid Lean4 code
- Be non-equivalent to the original statement
- Represent subtle but meaningful changes
- Output the complete code including all imports, definitions, and context
- Keep the original theorem name unchanged, only modify the theorem content/statement
- Preserve all other parts of the code (imports, helper definitions, etc.) exactly as given

Perturbation Methods for Reference (apply different ones):

1. Quantifier Modification

- Change $\forall$ to $\exists$ or vice versa
- Modify quantifier scope or order
- Add/remove quantifier constraints

2. Logical Operator Changes

- Switch $\wedge$ (and) with $\vee$ (or)
- Change $\rightarrow$ (implies) to $\leftrightarrow$ (iff) or vice versa
- Modify negation placement

3. Basic Operator / values Changes

- Change + to -
- Change the values of variables
- Swap two variables

4. Relational Operator Perturbation

- Change = to $\neq$, < to $\leq$, > to $\geq$, etc.
- Swap left and right sides of relations
- Modify strict vs non-strict inequalities

5. Structural Modifications

- Modify variable scoping or binding
- Alter type constraints or domains

6. Boundary Condition Changes

- Modify edge cases or boundary values
- Change inclusive/exclusive conditions

---

- Alter constraint ranges

Output Format:

Return exactly one JSON object with 6 perturbations:

```
{
    "original_analysis": "Brief analysis of the original statement'
    s key semantic components",
    "perturbations": [
        {
            "id": 1,
            "method": "Perturbation method used",
            "lean_code": "Modified Lean4 statement",
            "semantic_change": "Explanation of how the meaning
            changed"
        },
        {
            "id": 2,
            "method": "Perturbation method used",
            "lean_code": "Modified Lean4 statement",
            "semantic_change": "Explanation of how the meaning
            changed"
        },
        ...
        {
            "id": 6,
            "method": "Perturbation method used",
            "lean_code": "Modified Lean4 statement",
            "semantic_change": "Explanation of how the meaning
            changed"
        }
    ]
}
```

— Start of Original_Lean4Code —
{formal_statement}
— End of Original_Lean4Code —

## G.2 PROMPT FOR CONSISTENCY CHECK

---

**Prompt for Consistency Check**

Role: Lean & Formal Verification Expert

Input:

- Mathematical_Text: A math problem and its answer (no proof).
- Lean4Code: A Lean 4 theorem statement formalizing the problem. Proof is intentionally omitted (e.g., sorry).

Goal: Determine if the Lean theorem statement is an exact and faithful formalization of the mathematical problem. Do not evaluate or consider the answer or the proof. Your sole task is to verify the correctness of the formalization.

Evaluation Stages (All required):

1. Mathematical Text Analysis
Identify all structurally and semantically relevant components of the mathematical problem, including variables, types, quantifiers, constraints, logic structure, conclusion, and so on. The analysis should be based on the actual content of the text.

2. Lean4 Code Analysis (ignore proof part)
Extract all structurally and semantically relevant components from the Lean statement, including variables, types, conditions, quantifiers, constraints, the final claim, and so on. The analysis should reflect the actual content present in the Lean code.

3. Comparative Analysis
Check for exact correspondence between the math and Lean statements; you may refer to aspects like:
- Semantic alignment, logic structure, and quantifier correctness.
- Preservation of constraints and boundary assumptions.
- Accurate typing and use of variables.
- Strict adherence to Lean's specific syntactic and semantic rules in interpreting the Lean code.
- Syntactic validity and proper Lean usage (free from errors).
- Use of symbols and constructs without semantic drift.
- No missing elements, no unjustified additions, and no automatic corrections or completions.

4. Accuracy Confirmation
If correct: clearly confirm why all elements match.
If incorrect: list all mismatches and explain how each one affects correctness.

Note: While the analysis may be broad and open to interpreting all relevant features, the final judgment must be based only on what is explicitly and formally expressed in the Lean statement.

Do not consider or assess any part of the proof. Your judgment should be entirely about the accuracy of the statement formalization.

Output Format: Return exactly one JSON object:

```
{
    "reasons": "1. Mathematical Text Analysis: [...]
    2.  Lean4 Code Analysis: [...]
    3. Comparative Analysis: [...]
```

```
    4. Accuracy Confirmation: [...match confirmation or list of
    discrepancies...]",
    "is_assistant_correct":[Correct/Incorrect]"
}
```

## G.3 SYSTEM PROMPT FOR COLD START DATA SYNTHESIS

**System Prompt For Cold Start Data Synthesis**

You are an expert in mathematics and Lean 4.
Given a problem in natural language, your task is to convert the problem into valid Lean 4 statement with a header.
Your code should start with

```
import Mathlib
import Aesop
```

MANDATORY TOOL USAGE REQUIREMENT

You MUST use the provided tools to help verify your Lean4 statement. Tool calling is MANDATORY for EVERY code version you write. You CANNOT consider any code validated without explicit tool verification.

TOOLS:

- syntax_check: Call this function to verify whether a Lean4 statement can be compiled through Lean4 syntax, and return the compilation result.
- consistency_check: Call this function to verify whether the Lean4 statement that has passed the syntax_check is consistent with the original problem, return the responses of judge.

STRICT VERIFICATION WORKFLOW:

Step 1: Carefully analyze the problem statement and its mathematical meaning. Identify key components, relationships, and constraints. Write your initial Lean4 statement based on this analysis
Step 2: ALWAYS call syntax_check to verify your code compiles
Step 3: ONLY if syntax_check returns "pass": True, then call consistency_check
Step 4: If either check fails, carefully analyze the specific error messages. Identify the root causes of the issues. Then modify your code and RESTART the verification process
Step 5: REPEAT until BOTH checks pass successfully

HANDLING VERIFICATION FAILURES TIPS:

- If syntax_check fails: Analyze the errors, fix the issues, and IMMEDIATELY call syntax_check again with the corrected code
- If consistency_check fails: Review explanations, modify the statement to align with the problem, then restart verification with syntax_check
- After ANY code modification, no matter how minor, you MUST verify it with tools before proceeding

FINAL RESULT CRITERIA:

- A statement is considered correct ONLY when it explicitly passes BOTH syntax_check ("pass": True) AND consistency_check ("pass": True)
- Only after successful verification by BOTH tools should you present the code as the final result

- Do NOT declare completion without evidence of successful tool verification

LEAN 4 CODE REQUIREMENTS:

- The Lean 4 code must contain NO comments. All your reasoning, explanations, and analysis should be provided separately before presenting the code. The code itself should be clean and free of any embedded comments or documentation.
- All code must be compatible with Lean4 v4.15 syntax and features. Use only constructs and libraries available in this specific version, including proper notation, declaration formats, and namespace handling.
- Before you pass the Lean4 code as the arguments in tool call, you should write the code first. Remember add ":= by sorry" to the end of the statement.
- You should only output the statement in Lean 4 format as the final result. You should NOT complete the proof.

TOOL USE REQUIREMENTS:

- You MUST call the verification tools for EACH version of your code. Failure to call tools or skipping verification steps is NOT permitted. Never assume your code is correct without explicit tool verification.
- Always provide your reasoning, approach, and analysis before calling any verification tool. Explain what you're trying to achieve with the code and how it addresses the problem requirements.
- After receiving tool results, you must analyze them and explain your next steps before making additional tool calls.

## G.4 SYSTEM PROMPT FOR ATF INFERENCE

**System Prompt For ATF Inference**

You are an expert in mathematics and Lean 4. Your task is to convert natural language problems into valid Lean 4 formal statements (Compatible with Lean 4 v4.15).

Your code must begin with:

```
import Mathlib
import Aesop
```

You MUST use the provided tools to verify your Lean 4 statements:

- syntax_check: Verifies Lean 4 statement syntax
- consistency_check: Verifies that syntax-valid statements match the original problem

Verification workflow:

- Analyze the problem and create initial Lean 4 statement
- Call syntax_check to verify compilation
- If syntax check passes, call consistency_check
- If any check fails, analyze errors, modify code and restart verification
- Repeat until BOTH checks pass

