# OpenReview forum: "Autoformalizer with Tool Feedback"
_ICLR.cc/2026/Conference — Submitted to ICLR 2026_

### Official Review · Reviewer_ZNEc · 2025-10-30

**Soundness:** 3
**Presentation:** 3
**Contribution:** 3
**Rating:** 2
**Confidence:** 5

**Summary:**

ATF is a system that turns natural-language math problems into formal Lean 4 statements using feedback tools. It combines **syntax checks** from the Lean compiler and **semantic checks** from multiple LLM judges to iteratively refine results. Trained in three stages, ATF greatly improves both accuracy and consistency over previous models and releases a 750K-sample dataset (**Numina-ATF**) to support further research.

**Strengths:**

- The proposed framework, ATF, is clearly structured and the experimental results are reported in a generally comprehensible way.
- The topic itself is timely, and the authors make an effort to connect their work to recent trends in LLM-based reasoning and formal verification.

**Weaknesses:**

- I want to know what other tool calls, besides the **Syntax Check Tool**, can enhance autoformalization.
I doubt that there are many tools capable of surpassing **Lean** in terms of checking ability, so the paper should explore **Lean’s potential as a tool** more deeply.

- Lean is not good at performing numerical calculations, but I didn’t see you invoke any **calculator-related tools** in your framework.

- Please provide experiments on **benchmarks that require extensive numerical computation**.

**Questions:**

Please refer to Weaknesses.

---

> ### Author Response · Authors · 2025-11-19
> **# To Reviewer ZNEc**
>
> We appreciate the reviewer's expertise in formal languages. At the same time, we notice that your concerns differ significantly from those raised by the other reviewers, which suggests there may be a misunderstanding of our work. Automated theorem provers (ATPs) typically start from a formal statement/theorem and attempt to construct a proof. Our work focuses only on the statement autoformalization step of translating a user's natural-language query into an equivalent formal statement that an ATP can consume. For example:
> natural language query:
> “Show that the sum of the first 100 positive integers equals 5050.”
>
> formal statement:
> `theorem sum_first_100 : (∑ k : Fin 100, (k : Nat) + 1) = 5050 := by sorry`
>
> Widely accepted evaluation criteria for autoformalization require (1) the generated statement to be syntactically valid, and (2) the statement to be mathematically equivalent to the original query. The `syntax_check` tool is designed to enforce the first criterion, while the `consistency_check` tool targets the second. These tools are therefore aligned with the widely recognized standards for autoformalization quality. We adopted Lean as the target language because it is the lingua franca for state-of-the-art prover agents.
>
> Regarding **Weakness 1** on "exploring the potential of Lean as tools," the goal of `syntax_check` is precisely to help the model generate Lean statements that can pass the compiler. Prior work has explored some rule-based heuristics to aid autoformalization[1,2]. However, compiler feedback offers the most direct and universally applicable guidance on syntax errors, and we found that the model effectively leverages these messages to self-correct. Most recent works treat Lean compilation as the gold standard for syntax. While Lean could provide additional signals beyond syntax, those approaches demand bespoke engineering and suffer from limited generalization across variable names, domains, and goal types, which conflicts with the simplicity and broad applicability required for self-reflective reasoning in large language models. For example, StepFun-Formalizer[3] uses "beq" metrics[1] (in section 5.2) to facilitate majority voting. When using the same beq@k metric employed in StepFun-Formalizer, Goedel-v2-Formalizer-32B achieves beq@1=0.2350, which is significantly lower than Kimina-7B-Autoformalizer's beq@1=0.3957, clearly contradicting the actual capabilities of these models.
>
> In terms of **Weakness 2** about "calculator-related tools", as illustrated by the example above, autoformalization is more similar to a translation task that maps natural-language statements to their formal counterparts and does not involve any specific proofs. As such, calculator-style tools are unnecessary at the autoformalization stage. They are more relevant for the subsequent proof generation phase handled by dedicated prover models like kimina-prover or deepseek-prover.
>
> Regarding **Weakness 3** about "benchmarks involving numerical computation": Since autoformalization does not execute or verify numeric computations, our benchmark choice prioritized those used by powerful prover models. Please refer to the additional experimental evidence in the Overall Response for concrete results on those benchmarks.
>
> [1] Rethinking and Improving Autoformalization:
> Towards a Faithful Metric and a Dependency-Retrieval-Based Approach
>
> [2] Improving autoformalization using type checking
>
> [3] StepFun-Formalizer: Unlocking the Autoformalization Potential of LLMs through Knowledge-Reasoning Fusion

---

> > ### Comment · Reviewer_ZNEc · 2025-11-26
> >
> > Thank you for your rebuttal. I will maintain my score. Although the idea of leveraging compiler feedback is practical, it has already been adopted by many previous works[1，2 ], which limits the novelty of your approach.
> >
> > **Ref:**
> > [1] Aria: An Agent For Retrieval and Iterative Auto-Formalization via Dependency Graph.
> > [2] Lyra: Orchestrating Dual Correction in Automated Theorem Proving

---

### Official Review · Reviewer_7cvV · 2025-10-31

**Soundness:** 3
**Presentation:** 3
**Contribution:** 3
**Rating:** 6
**Confidence:** 4

**Summary:**

The paper proposes a model called ATF (Autoformalizer with Tool Feedback), which is designed to translate mathematical problems from natural language into formal statements. To this end, the authors design two types of tool feedback mechanisms: the first uses the Lean 4 compiler to check and correct the syntax of the generated formal statements, ensuring syntactic validity; the second adopts a multi-model voting approach to evaluate the semantic consistency of the generated results. When errors occur in the formalization results, the model can iteratively revise its outputs based on the tool feedback. To train ATF, the authors propose a three-stage training process: first, a “cold start” phase on synthetic data to teach the model how to use tools for correction; then an “expert iteration” phase to further improve the model’s capability through simulated expert feedback; and finally, a Direct Preference Optimization (DPO) phase to reduce ineffective modifications. In the experiments, ATF is evaluated on three mainstream benchmark datasets (FormalMath-Lite, ProverBench, and CombiBench), and the results show that ATF significantly outperforms the current best baseline, Goedel-V2-Formalizer-32B, in both syntactic validity and semantic consistency. The authors also release a synthetic dataset containing 750,000 formal statements (Numina-ATF) and conduct detailed human evaluations and ablation studies to verify the effectiveness of each component. Overall, this work demonstrates a new approach to significantly improve automatic mathematical formalization through tool feedback and provides new data resources.

**Strengths:**

- High innovation: For the first time, the paper introduces the use of Lean compiler outputs as a syntax verification tool and multi-model collective judgment as a semantic verification tool in the automatic formalization task, effectively combining the strengths of formal systems and large model reasoning.
- Significant experimental results: The proposed method substantially surpasses the current best approach (Goedel-V2-Formalizer-32B) on multiple popular benchmarks, including FormalMath-Lite, ProverBench, and CombiBench, with particularly notable improvements in semantic consistency metrics, demonstrating the effectiveness of the approach.
- Well-designed training process: The proposed three-stage training strategy—cold start, expert iteration, and DPO—progressively optimizes the model for different needs, enabling it to learn how to invoke tools and make reasonable corrections based on feedback, showing a thoughtful and well-structured design.
- Resource contribution: The authors release the Numina-ATF synthetic formalization dataset with 750,000 samples, providing the community with valuable resources for training and evaluation, which holds high practical value.
- Detailed analysis: The paper includes human evaluations and ablation studies, offering in-depth analysis of the roles of each component and the model’s scalability (such as extension effects in the inference stage), enhancing the credibility of the work. The writing is clear, and the figures are easy to read, making the contributions easy to grasp.

**Weaknesses:**

- Concerns about the reliability of the multi-model consistency tool: The paper relies on multiple large language models as “judges” to determine whether the generated statements are semantically consistent with the problems. However, the judgments made by LLMs may be unstable or biased, especially when it comes to subtle logical errors. Although the authors conducted human evaluations, the error rate and potential blind spots of the consistency checking tool remain unclear. It is recommended to further quantify or add verification mechanisms to ensure the accuracy of consistency feedback.
- Limited generalization ability and scope: The current tool feedback is based on the Lean 4 compiler. If mathematical problems need to be formalized in other languages (such as Isabelle or Coq) or in different versions of Lean, the current approach may not be directly applicable. The authors mention the differences between Lean versions, but there is insufficient study on the adaptability of the method. Future work could explore the method’s transferability across different formal systems or introduce language-agnostic tool interfaces.
- Training and inference overhead: The three-stage training and multi-round feedback mechanisms of ATF increase computational complexity. In particular, during inference, the repeated invocation of the compiler and multi-model judgments may lead to slower inference speed and higher resource consumption. The paper does not discuss efficiency in detail. In practical applications, fast response is also important, and it would be helpful for the authors to specify the model’s inference cost and latency, as well as its performance under limited computational resources.
- Interpretability and failure analysis: Although the paper provides overall performance improvement data, it lacks an in-depth analysis of failure cases. For example, it remains unclear what types of problems ATF still struggles to formalize, or when tool feedback fails to correct the output. A detailed analysis of failure cases would help reveal the limitations of the approach and potential directions for improvement.

**Questions:**

- The paper mentions using multiple large language models (LLMs) as “judges” for semantic consistency verification. However, I only observed the use of **QWQ-32B** and **Qwen3-32B** in the main text. Could the authors clarify whether these are the only LLMs employed in the consistency check, or if other models were also used but not explicitly mentioned in the paper?
- In Table 1, the results do not appear to show a clear advantage of the *Ensemble Vote* method compared to using a single LLM as the judge. I would recommend adding a new evaluation metric — **Accuracy** — to the table, which would provide a more intuitive comparison and make it easier to quantify the improvement brought by the ensemble voting method.
- Since ATF requires multiple rounds of tool calls for iterative correction during inference, does this lead to a significant computational overhead? How does the actual inference speed compare to conventional one-shot formalization models? Moreover, have the authors considered strategies such as reducing the number of iterations or parallelizing the process to enhance scalability for large-scale mathematical libraries?
- In Section 5.2 (“Tool Analysis”), the paper states: “As shown in Figure 5, the number of tool calls varies by dataset; CombiBench requires the highest average number of tool invocations (8.35) due to its combinatorial complexity, while FormalMath-Lite requires fewer attempts (3.19).”However, I was unable to locate the corresponding values (8.35 and 3.19) in Figure 5. Similarly, the sentence *“ProverBench is an exception where consistency checking (66.34%) outperforms syntax checking (61.65%)”* cites values that also do not appear in the figure. Could the authors verify whether these numbers are accurate or possibly correspond to an earlier version of the figure?
- In Table 4 (ablation study), the improvement brought by adding the DPO training stage over the *Expert Iteration* stage alone appears rather marginal (around 1% increase). Could the authors elaborate on whether the DPO stage provides additional benefits beyond accuracy improvement, such as better stability, generalization, or robustness in handling ambiguous formalization cases?
- The experiments report strong results on **FormalMath-Lite**, **ProverBench**, and **CombiBench**. However, other widely used benchmarks for formalization tasks include **MiniF2F** and **ProofNet**. Could the authors share any results or observations on these datasets, or discuss potential challenges in applying ATF to them?

---

> ### Author Response · Authors · 2025-11-19
> **# To Reviewer 7cvV, about Weakness**
>
> 1. Weakness 1 about "reliability of consistency tool"
>
>     This concern is valid as we have also observed that current formalizers can generate subtle logical errors. As mentioned in Section 3.1.2, when designing the consistency tool, we specifically constructed a benchmark to evaluate the reliability of LLM-as-judge in this regard. The detailed construction of the benchmark can be found in Appendix A.2. When generating negative perturbations, we specified the following categories covering various types of logical errors:
>     1. **Quantifier Modification**
>     2. **Logical Operator Changes**
>     3. **Basic Operator / Values Changes**
>     4. **Relational Operator Perturbation**
>     5. **Structural Modifications**
>     6. **Boundary Condition Changes**
>
>     Therefore, we assume this multi-LLMs-as-judge approach is relatively reliable.
>
> 2. Weakness 2 about "limited scope"
>
>     We chose Lean 4 because several mainstream Prover and Autoformalizer models use Lean 4 as the formalization language. Migrating the syntax tool to other formalization languages is more of an engineering issue and can be left for future work.
>
> 3. Weakness 3 & Question 3 about "Training and inference overhead".
>
>     We believe the training overhead is not a concern because the compared baselines generally also undergo heavy training during the training phase. In contrast, inference overhead is more important. Tool calls inevitably incur additional overhead. The overhead from the syntax tool due to grouped Lean 4 execution optimization is negligible compared to model inference. The main additional overhead comes from consistency tool calls. For execution efficiency, we made two optimizations in the rules: 1. The syntax tool must be executed before the consistency tool, and 2. When the first judge rejects, we directly return the result as feedback. At the engineering level, we also implemented an asynchronous architecture for tools and model inference (refer to the submitted code), minimizing the delay caused by tool execution as much as possible. Regarding the inference overhead of the formalizer model, we set revisions < 4 to ensure that the inference path length remains consistent with Goedel V2, ensuring a fair comparison.
>
> 4. Weakness 4 about "failure analysis"
>
>     We selected CombiBench (which includes more human-checked failed cases) to conduct a simple failure analysis. We found that most failures can be attributed to over-simplifying the transformation from mathematical descriptions to formal statements. A brief but representative example is shown below.
>
>     Natural language query:
>
>     How many ways can a teacher select a group of 6 students to sit in the front row if the class has 13 students? Prove that the answer is 1716.
>
>     Generated formal statement:
>
>     ```lean
>     import Mathlib
>     import Aesop
>
>     theorem my_favorite_theorem :
>         (13 * 12 * 11 * 10 * 9 * 8) / (6 * 5 * 4 * 3 * 2 * 1) = 1716 := by
>       sorry
>     ```
>
>     Strictly, the theorem should be formalized as `Nat.choose 13 6 = 1716`. Although the numeric expression `(13 * 12 * 11 * 10 * 9 * 8) / (6 * 5 * 4 * 3 * 2 * 1)` is equivalent in value to `Nat.choose 13 6`, it over-simplifies the combinatorial reasoning into pure arithmetic manipulation, and such formalizations are manually judged as incorrect. We will conduct a more comprehensive failure analysis on cases from additional datasets in the revised version.

---

> > ### Author Response · Authors · 2025-11-19
> > **# To Reviewer 7cvV, about Question**
> >
> > 5. Question 1 about "selection of judges"
> >
> >     In this work, we only used QwQ and Qwen3-32B as judges (details are described in Appendix A.2). In fact, we also evaluated other open-source models as judges, but QwQ and Qwen3-32B performed best (for example, Qwen2.5-Coder-32B-Instruct only achieved an FPR of 15.4). Considering both deployment costs and effectiveness, we only used these two models.
> >
> > 6. Question 2 about "metrics of llm-as-judges"
> >
> >     The accuracy metrics for different judges are as follows:
> >     - QwQ: 0.8347
> >     - Qwen3-32B: 0.8358
> >     - Ensemble Vote: 0.8236
> >
> >     Note that because the benchmark construction results in more negative samples than positive samples, the accuracy metric cannot accurately reflect the performance of the judges. Even though the ensemble vote effectively imposes a stricter acceptance threshold, the accuracy does not show a noticeable decline. We focus on the FPR metric, which directly measures how many of the accepted samples are actually incorrect.
> >
> > 7. Question 4 about "confusion of figures"
> >
> >     We apologize for the confusion caused by the figures. We have carefully checked and found that the inconsistency in the average number of tool calls is due to invalid tool calls. The tool analysis is based on cases with revision times < 16 in Section 5.1. During the evaluation phase, there is still a small probability that model tool calls fail, especially in trajectories with a larger number of tool call rounds. In such cases, we return an invalid tool feedback (as implemented in the code). When plotting, we only plot valid tool calls, but we do not exclude this part when doing statistical analysis. The inconsistency in tool call success rates is due to decimal place rounding issues. In the figure, 66.34% was approximated to 66.3%, and 61.65% was approximated to 61.7%. We will make corrections in the revised version to avoid reader confusion.
> >
> > 8. Question 5 about "additional benefits of DPO phase"
> >
> >     Due to the substantial training volume in the expert iteration phase, the DPO stage shows limited performance improvements. The motivation for DPO stems from our observation that models after expert iteration often exhibit repetitive errors during reasoning (e.g., the syntax tool repeatedly reporting errors on the same variable definition). DPO is designed to encourage the model to make more effective corrections. We therefore compared the average number of tool calls (counting only valid tool calls) across all valid trajectories during evaluation before and after the DPO phase:
> >
> >     |                     | combibench | formalmath-lite | proverbench |
> >     |---------------------|------------|-----------------|-------------|
> >     | before dpo          | 9.17        | 3.20             | 4.22        |
> >     | after dpo           | 8.21       | 3.17             | 3.84         |
> >
> >     The results show that after DPO, the number of tool calls in successful trajectories on CombiBench decreased significantly (from 9.17 to 8.21), while the change on FormalMath-lite was minimal (from 3.20 to 3.17). This observation may provide a possible explanation for why DPO shows more pronounced performance improvements on CombiBench compared to FormalMath-lite in the ablation study.
> >
> >
> >
> > 9. Question 6 about "limited evaluations"
> >     Please refer to the experimental results presented in the "Overall Response" above.

---

### Official Review · Reviewer_A4wN · 2025-10-31

**Soundness:** 2
**Presentation:** 3
**Contribution:** 2
**Rating:** 6
**Confidence:** 2

**Summary:**

This paper presents Autoformalizer with Tool Feedback (ATF), a framework that integrates Lean 4 compiler feedback and multi-LLM semantic evaluation to improve mathematical autoformalization. Through three-stage training—cold start, expert iteration, and DPO—ATF learns effective tool usage and revision strategies. Experiments on FormalMath-Lite, ProverBench, and CombiBench show significant gains over prior systems like Goedel-V2 and StepFun-Formalizer, particularly in semantic consistency.

**Strengths:**

1. The paper clearly identifies two key bottlenecks in current autoformalization models—syntactic errors and semantic drift—and systematically addresses both through tool feedback.
2. The experiments show substantial improvements in syntactic validity and semantic consistency, further validated by human evaluation, demonstrating a strong correlation with human judgment.
3. The system is thoughtfully designed, featuring grouped execution and expert iteration mechanisms that enable efficient syntax checking and progressive tool learning.
4. The paper is well-written and comprehensive, with detailed appendices and clear figures that effectively illustrate the model’s iterative reasoning and tool interaction process.

**Weaknesses:**

1. The paper introduces a meaningful but moderately novel approach by systematizing tool feedback specifically for autoformalization
2. Training and evaluation datasets are all derived from the Numina ecosystem; although similarity-based decontamination (cosine < 0.8) is performed, stronger guarantees against overlap would make the results more convincing. Include one external dataset (e.g., MiniF2F or PutnamBench) or a stricter decontamination threshold.

**Questions:**

Please refer to the Weakness section.

---

> ### Author Response · Authors · 2025-11-19
> **# To Reviewer A4wN**
>
> 1. Weakness 1 about "moderate novelty".
>
>     This work is motivated by the observation that training prover models requires a large number of formal statements, while existing autoformalizer models fail to produce high-quality statements. We primarily focus on designing two tools that can effectively improve the quality of formal statements, placing greater emphasis on practicality to help facilitate subsequent ATP-related research. Future work will expand the scope of applications, such as formalizing complex mathematical proof processes.
>
> 2. Weakness 2 about "risk of contamination".
>
>     In autoformalization, natural language queries are not overly complex, and embeddings can effectively capture the semantic information. Therefore, the decontamination threshold of cosine similarity < 0.8 adopted in the paper is already a relatively strict standard. The evaluation on the additional datasets also employed the same decontamination standard, and the results further validate the effectiveness of ATF.

---

### Author Response · Authors · 2025-11-19
**Overall Response to All Reviewers**

Dear Reviewers,

We sincerely thank all reviewers for their thorough and constructive reviews. We greatly appreciate their recognition of the significance and the novelty of this work, our contribution to the ATP community.

We have carefully considered all concerns raised and hope we have adequately addressed each reviewer's questions in the individual responses below.

Regarding the common suggestion raised by all reviewers to expand our evaluation to additional benchmarks, we provide a unified response here:

We initially selected FormalMath-Lite, ProverBench, and CombiBench to follow the experimental setup in StepFun-formalizer, ensuring fair and direct comparison among the most recent fromalizers. To better support the conclusions in this paper, we have conducted additional experiments on **four widely-used automated formal proving benchmarks**: MiniF2F, ProofNet, MathOlympiadBench, and PutnamBench, keeping all experimental parameters consistent with those reported in the paper. The results are presented in the table below:

| **Model** | **MiniF2F** || **ProofNet** || **MathOlympiadBench** || **PutnamBench** ||
|----------------------------|:------:|:------:|:------:|:------:|:------:|:------:|:------:|:------:|
| | SC | CC | SC | CC | SC | CC | SC | CC |
| _Pass@1_ |||||||||
| Kimina-7B | 93.36| 75.37 | 52.26 | 30.70 | 26.58 | 10.47 | 73.68 | 31.72 |
| StepFun-7B | 93.07 | 80.61 | 44.30 | 30.32 | 74.56 | 45.69 | 57.73 | 27.16 |
| Goedel-V2-8B | 97.46 | 87.50 | 69.46 | 57.15 | 84.42 | 55.58 | 79.47 | 49.10 |
| StepFun-32B | 96.11 | 84.55 | 54.25 | 38.71 | 79.94 | 49.86 | 62.33 | 35.73 |
| Goedel-V2-32B | 97.17 | 89.96 | 68.01 | 58.55 | 84.89 | 58.17 | 80.39 | 53.59 |
| ATF-32B (Ours) | 99.15 | 97.31 | 69.56 | 62.16 | 91.44 | 81.70 | 88.76 | 76.14 |
| _Pass@8_ |||||||||
| Kimina-7B | 99.43 | 94.39 | 78.82 | 56.72 | 62.78 | 34.61 | 94.05 | 57.92 |
| StepFun-7B | 99.02 | 97.01 | 67.26 | 54.41 | 91.19 | 71.4 | 91.57 | 65.56 |
| Goedel-V2-8B | 99.34 | 98.52 | 83.12 | 76.83 | 96.28 | 83.56 | 95.53 | 80.90 |
| StepFun-32B | 99.63 | 97.99 | 74.95 | 66.02 | 92.72 | 76.33 | 82.59 | 63.48 |
| Goedel-V2-32B | 99.59 | 99.10 | 85.54 | 77.20 | 96.36 | 86.92 | 95.39 | 86.06 |
| ATF-32B (Ours) | 99.80 | 99.39 | 91.76 | 85.52 | 99.26 | 96.15 | 98.58 | 95.57 |
| _Pass@16_ |||||||||
| Kimina-7B | 99.59 | 96.72 | 82.80 | 61.83 | 70.00 | 43.06 | 96.12 | 64.60 |
| StepFun-7B | 99.18 | 97.95 | 70.97 | 58.60 | 93.61 | 76.11 | 94.41 | 75.16 |
| Goedel-V2-8B | 99.59 | 98.77 | 86.56 | 79.03 | 97.50 | 87.78 | 97.20 | 87.11 |
| StepFun-32B | 100.00 | 98.77 | 78.49 | 70.43 | 94.44 | 80.56 | 85.56 | 70.19 |
| Goedel-V2-32B | 99.59 | 99.59 | 88.17 | 80.65 | 97.22 | 91.39 | 97.05 | 70.19 |
| ATF-32B (Ours) | 100.00 | 99.59 | 94.62 | 89.25 | 99.72 | 97.78 | 99.22 | 97.52 |

The results show that ATF-32B still achieves the best results on every dataset. Notably, on MathOlympiadBench and PutnamBench, ATF achieves significant improvements, demonstrating the effectiveness of revisions based on feedback.

---

### Author Response · Authors · 2025-11-28
**Hope this comment helps AC**

We add this comment to prevent potential confusion caused by rating rollback. Reviewer ZNEc initially misunderstood certain technical aspects of our work, leading to a lower initial score that diverged from other reviewers' assessments. Through explanations in our response, the reviewer acknowledged our method's practical utility and implementation value, updating rating to 6 as the corrected initial rating. After considering the rebuttal content, they explicitly claim to maintain rating = 6 as the final score.  We hope this comment can assist AC in better evaluating the value of our contribution.

---

### Meta-Review · Area_Chair_ZGcY · 2026-01-05

**Summary:**

This paper introduces an approach of autoformalizer with tool feedback (ATF) to address the syntactic validity and semantic consistency problems in Lean 4 autoformalization. The framework of ATF consists of three training stages: cold start, expert iteration, and DPO. Evaluations are on FormalMath-Lite, ProverBench, and CombiBench for syntax check and consistency check. The results show that ATF consistently outperforms the compared formalizers (Kimina-Autoformalizer-7B, StepFun-Formalizer-
7B/32B, and Goedel-V2-Formalizer-8B/32B).

**Reviewer Concerns:**

The reviewer concerns that are addressed:
* Failure analysis (Reviewer 7cvV) -> by providing a case from CombiBench.
* Provide experiments on benchmarks that require extensive numerical computation (Reviewer ZNEC) -> more experiments on MiniF2F, ProofNet, MathOlympiadBench, and PutnamBench
* Not including calculator-related tools in the framework (Reviewer ZNEC) -> distinguishing autoformalization as a translation phase from the subsequent proof generation phase.

Also, other reviewer concerns are partially addressed:
* Moderate novelty (Reviewer A4wN)
* Risk of data contamination (Reviewer A4wN)
* The reliability of the multi-LLMs-as-judge approach (Reviewer 7cvV)
* Training and inference overhead (Reviewer 7cvV)
* Limitations of exploring the potential of Lean as a tool (Reviewer ZNEC)

**Reviewer Scores:**

In general, the author's responses address part of the reviewers' concern. Reviewer ZNEC may raise his/her score. However, the overall ratings are marginal.

---

### Decision · Program_Chairs · 2026-01-26

Reject